# Improving CNN-Based Mitosis Detection through Rescanning Annotated Glass Slides and Atypical Mitosis Subtyping

**Rutger H.J. Fick**                                         FICK.RUTGER@GMAIL.COM
*Tribun Health*

**Christof A. Bertram**                          CHRISTOF.BERTRAM@VETMEDUNI.AC.AT
*University of Veterinary Medicine Vienna*

**Marc Aubreville**[*]                                      MARC.AUBREVILLE@THI.DE
*Technische Hochschule Ingolstadt*

**Editors:** Accepted for publication at MIDL 2024

## Abstract

The identification of mitotic figures (MFs) is a routine task in the histopathological assessment of tumor malignancy with known limitations for human observers. For a machine learning pipeline to robustly detect MFs, it must overcome a variety of conditions such as different scanners, staining protocols, tissue configurations, and organ types. In order to develop a deep learning-based algorithm that can cope with these challenges, there are two obstacles that need to be overcome: obtaining a large-scale dataset of MF annotations spread across different domains of interest, including whole slide images (WSIs) exhaustively annotated for MFs, and using the annotated MFs in an efficient training process to extract the most relevant features for classification. Our work attempts to address both of these challenges and establishes an MF detection pipeline trained solely on animal data, yet competitive on the mixed human/animal MIDOG22 dataset, and, in particular, on human breast cancer. First, we propose a processing pipeline that allows us to strengthen the *true* scanner robustness of our dataset by physically rescanning the glass slides of annotated WSIs and registering MF positions. To enable the use of such rescans for training, we propose a novel learning paradigm tailored for labels that match partially, which allows to account for ambiguous MF positions in the rescans caused by spurious, suboptimal fine-focus on potential MFs by the scanner. Second, we demonstrate how a multi-task learning approach for MF subtypes, including the prediction of atypical mitotic figures (AMFs), can significantly enhance a model's ability to distinguish MFs from imposters. Our algorithm, using a standard object detection pipeline, performs very competitively with an average test set F1 value across five runs of 0.80 on the MIDOG22 training set. We also demonstrate its ability to stratify overall survival on the TCGA-BRCA dataset based on mitotic density, though it falls short of reaching significance in stratifying survival based on AMFs.

**Keywords:** Whole Slide Imaging, Mitosis Detection, Atypical Mitosis Subtyping, Deep Learning

---

[*] corresponding author

## 1. Introduction

The identification of morphological structures of dividing cells, known as mitotic figures (MFs), is a routine task in the histopathologic assessment of tumor malignancy (Donovan et al., 2021). MFs are morphologically heterogeneous and can be either normal or atypical. Due to their morphological complexity, MFs are prone to be missed, or mistaken for other apoptotic or necrotic cell structures during manual assessment, resulting in significant inter-rater disagreement (Meyer et al., 2005; Malon et al., 2012; Veta et al., 2016). For these reasons, and also to reduce the workload of pathologists, automatic MF identification is a well-established computer vision task. The success of recent methods, which show an accurate and reproducible detection of MFs on regions of interests (ROIs) was, to a large degree, fueled by the availability of highly diverse and sufficiently large datasets, such as AMIDA13 (Veta et al., 2015), TUPAC16 (Veta et al., 2019), MIDOG21 (Aubreville et al., 2023b) and MIDOG22 (Aubreville et al., 2023c). Common to those datasets, is the limitation that only annotations for a pre-selected ROI exist, that is typically selected from the area of highest MF density within the tumor. This restricts the data diversity of those datasets, in particular by not including areas that may contain cells such as aptotic or necrotic cells, that can be easily mistaken for MFs and thus can be considered hard examples for the pattern recognition problem. Practically, this means that the application to the complete whole slide image (WSI) can become an out-of-distribution problem for detectors solely trained on ROIs. However, annotation of whole tumor sections is a labor-intensive task. In the field of canine histopathology, two notable datasets have been made publicly available, that, combined, provide annotations for more than fifty thousand mitotic figures, collected from 53 tumor specimens (Bertram et al., 2019; Aubreville et al., 2020). These annotations, however, have been provided for slides scanned with a single scanner only, limiting domain generalization across scanners. Annotation of a wide range of WSIs acquired on multiple scanners is infeasible due to the high cost of skilled labor for this task. As a cost-effective alternative, our first contribution in this study is a training paradigm that allows us re-use the efforts to exhaustively annotate these WSI by rescanning their physical glass slides and filtering spurious ambiguous MFs in the rescans appropriately.

Recently, it has also been reported that subtyping of MFs into normal and atypical mitotic figures (AMFs), which indicate an aberration of the normal chromosome separation process resulting in genetic alterations, might be relevant for the calculation of additional, prognostically relevant criteria in the assessment of breast cancer (Ohashi et al., 2018; Lashen et al., 2022). Initial work on the automatic subtyping of MFs (Aubreville et al., 2023a) has found this to be a challenging task, additionally restricted by a low inter-rater agreement.

In this study, we introduce a pipeline for detecting and subclassifying MFs in multi-organ, multi-scanner, multi-species, and fully WSI-based settings, demonstrating that sub-classifying mitoses in multi-task learning significantly enhances the performance of MF detection. Our approach, which was trained on a diverse set of fully annotated canine samples scanned with seven different systems, uses a robust training objective that is unaffected by out-of-focus mitoses in rescans. Our main contributions can be summarized as:

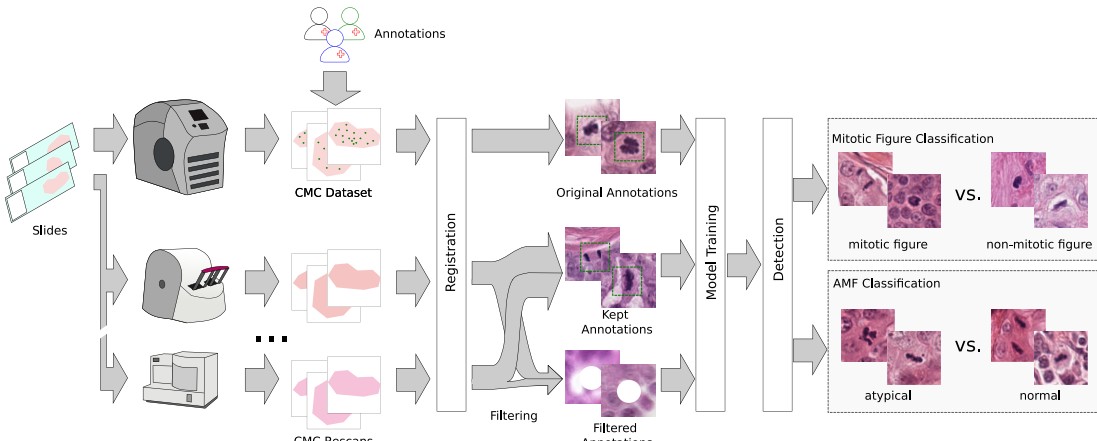

Figure 1: Overview: We register rescanned glas-slides of canine breast cancer and employ a novel filtering paradigm to disregard unrecognizable mitotic figures (MF). We additionally employ multi-task learning with atpical MF classification.

- We introduce a tailored training objective to counteract focusing artifacts commonly occurring when rescanning slides, allowing for the training of MF detectors with registered slides digitized by multiple scanners.

- We show that adding a atypical classification subtask regularizes the mitotic figure detection task, leading to consistenly better results.

- We show that our detector delivers both SOTA performance on the MIDOG22 challenge set and also stratifies survival on the BRCA-TCGA breast cancer WSI dataset.

## 2. Materials and Methods

In this section we first describe our MF dataset in section 2.1, followed by our WSI rescan filtering paradigm in section 2.2, and finally our model training approach in section 2.3.

### 2.1. Mitosis Dataset

A clinically usable MF detection algorithm must be robust to the varying tissue composition and quality conditions found in a WSIs, as well as to domain shifts due to different scanner and organ domains. Our dataset is designed to provide robustness to these challenges:

**WSI Robustness** We include the canine mammery carcinoma (CMC) dataset (Aubreville et al., 2020), which consists of 21 WSIs exhaustively annotated for MFs.

**Scanner Robustness** We rescanned a selection of the original CMC glass slides with 6 other scanners. We then transferred the MF annotations from the original WSI to the rescans using a WSI-level registration algorithm (Marzahl et al., 2021). To remove blurred, out-of-focus or otherwise missing MFs in the rescans we designed a custom filtering approach, which we detail in section 2.2.

| Dataset | Scanner | Resolution | No. cases | No. mitotic figures | No. atypical mitotic figures |
|---|---|---|---|---|---|
| CMC original | Aperio Scanscope | $0.25\,\frac{\mu m}{px}$ | 21 WSI | 14154 | 1533 |
| | Hamamatsu HT2.0 | $0.23\,\frac{\mu m}{px}$ | 18 WSI | 9724/12694 | 1039 |
| | Hamamatsu S360 | $0.23\,\frac{\mu m}{px}$ | 18 WSI | 8987/12694 | 940 |
| CMC re-scanned glass slides | 3DHistech Scan II | $0.25\,\frac{\mu m}{px}$ | 18 WSI | 9840/12694 | 1092 |
| | 3DHistech Flash III | $0.12\,\frac{\mu m}{px}$ | 18 WSI | 8660/12694 | 950 |
| | Philips SG360 | $0.25\,\frac{\mu m}{px}$ | 4 WSI | 4393/5605 | 663 |
| | Olympus VS200 | $0.25\,\frac{\mu m}{px}$ | 1 WSI | 1088/1343 | 204 |
| multi tumor | 3DHistech Scan II | $0.25\,\frac{\mu m}{px}$ | 159 ROI | 4670 | 400 |

Table 1: Overview of our training dataset, indicating the number of all and atypical MFs. For the rescans, we show the number of MFs remaining *after* our filtering scheme.

**Organ Robustness** We acquired a secondary dataset of 159 WSIs, covering 17 different cancer types throughout different organs and animal species. As WSI and scanner robustness is obtained from the previous two datasets, here we only selected and annotated one ROI per WSI for MFs.

We provide an overview of our composite dataset in Table 1. Our hold-out test set for MF detection is the MIDOG22 training dataset (Aubreville et al., 2023c), which consists of mitotic figure annotations in regions of interest originating from multiple species, organs and scanners. To allow for our secondary MF subtyping strategy, a trained pathologist also subtyped all annotated MFs into normal MFs or AMFs. We find that AMFs are *rare*, representing only about 10% of all mitotic figures. Our atypical MF test set consists of the MIDOG21 training set (Aubreville et al., 2023b), whose mitotic figures were similarly subtyped into atypical and normal mitoses in previous work (Aubreville et al., 2023a).

### 2.2. Training paradigm for rescanned slides

Rescanning the original CMC glass slides and transfering the MF annotations to different scanners allows us to re-use the massive effort that was done to exhaustively annotate the over 14K MFs in the original scanner domain by 2+1 experts (Aubreville et al., 2020). However, while the rescanned WSI *should* represent the same tissue as the original, the scanning process itself is not perfect and can result in MF annotation errors in the rescanned scanner domains. False annotations occur for various reasons: 1) MFs, which were in-focus in the original WSI are out-of-focus (OOF) in the rescan; 2) the scanning area has device-dependent limitations, leading to parts of the tissue not being scanned in areas of registered MFs; 3) the scanner's stitching algorithm can locally cut out mitotic figures if they live at the intersection between two stitched patches. Missing annotations in the rescan occur when MFs that were not visible in the original scanner domain are now in-focus in the rescan, but are not annotated. We show examples of false annotations in Figure 2. Given the scale of the MF annotations in our CMC rescan dataset, it is infeasible to manually verify whether each registered MF is *still* interpretable as such in the rescan. To clean our dataset for such errors without expert intervention, we propose a filtering approach based on the premise that false annotations MFs in rescanned images are caused by spurious scanner artifacts that have a random distribution. We trained 10 different classification architectures on all scanner domains and merge them into one big ensemble. Knowing that this ensemble will

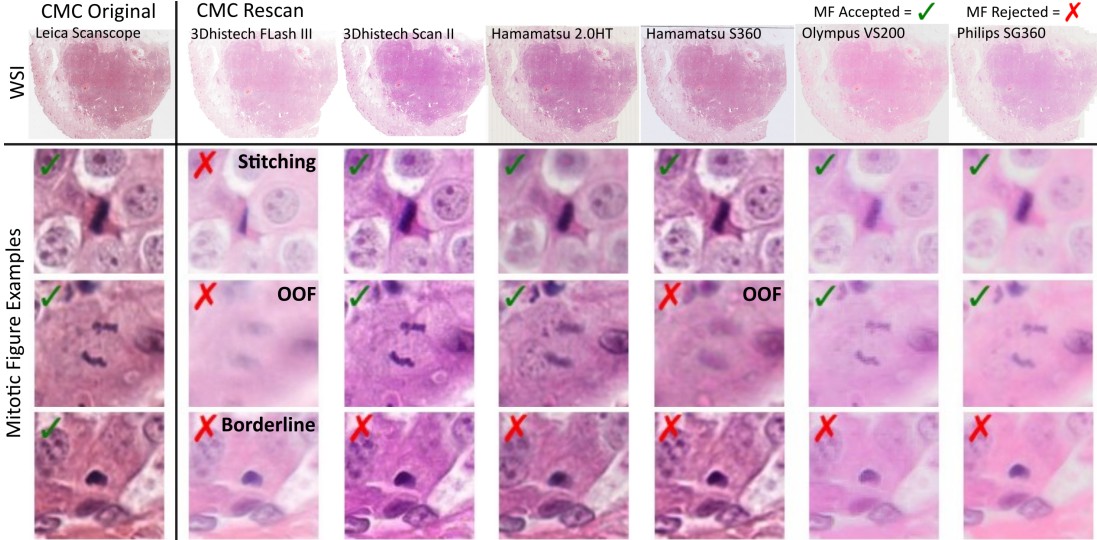

Figure 2: The first row illustrates the original and rescanned WSIs. Subsequent rows showcase examples of MFs from the original slide, followed by registered MFs from the rescans. The filtering process acceptance or rejection for training is represented by a green checkmark or red cross, respectively. MFs were rejected due to stitching artifact, being out of focus, and borderline morphology.

be reasonably robust to different scanners *and label noise*, we apply this ensemble to the rescanned WSI and apply a conservative threshold to mask any uncertain MF. During the detector training, when such masked objects reside in the (larger) patch that is sampled, we opt to draw white circles over the suppressed MFs so the training ignores uncertain objects. We chose to not use this approach to pseudo-label possible missed MFs annotations in the rescans as we felt we might end up validating MF lookalikes.

### 2.3. Mitosis Detection Pipeline

Our detection pipeline for MF detection is combination of a detector network to propose MF candidates, and an ensemble of classifiers that refines the selected candidates, which has proven to be successful in many MF approaches (e.g. Li et al., 2018; Piansaddhayanaon et al., 2023). Our detector network is a YOLOR-D6 (Wang et al., 2021) and our ensemble of two classifiers consists of a DenseNet201 (Huang et al., 2017) and an EfficientNetB4 (Tan and Le, 2019). We chose these architectures heuristically as those contributing most to the MF detection F1 score. For the classifier decoder heads, in addition to the standard binary MF/non-MF head we add a secondary head for binary normal/atypical MF classification, realized by a DenseNet201 network. We resampled all WSIs to $25\frac{\mu m}{px}$ and determined all thresholds on the validation set.

**Detector Training and Data Sampling** Since we have such a large-scale, diverse and *unbalanced* dataset for organ types (see Table 1 and Supplementary Table S1), it is important to guide the model training to not focus only on the majority groups. Therefore,

we adjusted the sampling probabilities to 50% from the base CMC dataset, and 25% each from the rescanned CMC and multi-organ datasets, and sampling with equal probability between the subgroups of those sets. We maintained a 50-50 split between MF annotations and negative annotations while maintaining the natural atypical/normal mitosis distribution. This strategy encourages model robustness w.r.t. different scanners in the rescanned dataset and to varying organs/cancer types in the multi-organ ROIs while leveraging the trustworthy annotations of the CMC base dataset. We trained on $1024 \times 1024$ px images until convergence was observed using the F1 score on the validation set, for which we assigned two cases from the CMC sets and 17 ROIs (one per tumor type) from the multi tumor set. We used an open-source library for HE-based data augmentation (Faryna et al., 2021).

**Classifier Training and Data Sampling** We adopted a similar domain sampling distribution for our classifier networks as for the detector. To train these networks, we used false positives generated by the trained detector model as negative examples, effectively utilizing the network as refining model (Li et al., 2018). We optimized the ensemble weighting and decision threshold for each validation step using grid search, using the best performing model per architecture (on the validation set) out of five runs. The second classifier head only handles the normal or atypical MF subtyping. For this reason, we created a separate dataloader that samples *only* MFs and samples normal and atypical mitoses equally. Within each category, we sampled the subtypes equally. Given that the primary head is used for early stoppping, it is not expected that the second head is optimal at the same time. For this reason, after convergence of the MF classification head, we froze the encoder and trained the secondary head until convergence.

## 2.4. Survival prediction on TCGA-BRCA

To investigate the WSI-based performance, we evaluated our detection pipeline on the breast cancer cohort (BRCA) of The Cancer Genome Atlas (TCGA) project. Following the diagnostic grading process of breast cancer (Fitzgibbons and Connolly, 2023), we selected the area with the highest mitotic count (MC), using a circular field of view spanning $2\,mm^2$. Since the classification head for atypical/normal shows only average accuracy, we selected only the predictions with the highest confidence ($p > 0.9$) for either class, reducing the number of detected MFs utilized for the next step by 42.84%. We then calculated the AMF to normal MF (NMF) ratio per case, similar to (Lashen et al., 2022). Due to the expected high level of inaccuracies in the AMF/NMF decision, we elected to calculate the ratio not only per ROI but also per WSI in an effort to decrease the SNR of the metric. We then fitted the survival data provided in the dataset to find the threshold which provided the strongest predictive value, as indicated by the p value of the Cox linear hazard model and fitted a Kaplan Meier estimator on this optimal threshold.

## 3. Results

Comparing the pure classification performance (second stage of the approach) for different network architectures, we find that adding the atypical subtyping task consistently improves the performance (see Table 2). We find this to also hold true in the end-to-end performance

| | Mitosis Detection F1 | | |
|---|---|---|---|
| Classifier Model | w/o filtering, w/o MF subtyping | w/ filtering, w/o MF subtyping | w/ filtering, w/ MF subtyping |
| DenseNet201 (Huang et al., 2017) | $0.72 \pm 0.022$ | $0.74 \pm 0.011$ | $\mathbf{0.77} \pm 0.025$ |
| EfficientNet B4 (Tan and Le, 2019) | $0.69 \pm 0.030$ | $0.72 \pm 0.013$ | $\mathbf{0.75} \pm 0.020$ |
| ResNet50 (He et al., 2016) | $0.70 \pm 0.016$ | $0.71 \pm 0.020$ | $\mathbf{0.74} \pm 0.014$ |

Table 2: Ablation study of both contributions on the second stage ($\mu \pm \sigma$ over five runs).

| | Dataset | Scanner | Primary Task F1 Mitosis Detection wo/ MF subtyping | w/ MF subtyping | Secondary Task F1 Mitosis Subtyping Perfect Primary | End2End |
|---|---|---|---|---|---|---|
| validation | Canine CMC Orig. | Aperio Scanscope | 0.774 | **0.811** | 0.815 | 0.696 |
| | Canine CMC Rescan | Hamamatsu HT2.0 | 0.783 | **0.808** | 0.771 | 0.649 |
| | | Hamamatsu S360 | 0.816 | **0.848** | 0.789 | 0.666 |
| | | 3DHistech Scan II | 0.781 | **0.837** | 0.808 | 0.697 |
| | | 3DHistech Flash III | 0.808 | **0.832** | 0.806 | 0.650 |
| | | Philips SG360 | 0.805 | **0.850** | 0.796 | 0.671 |
| | | Olympus VS200 | 0.792 | **0.840** | 0.774 | 0.647 |
| | Animal MultiTumor | 3DHistech Scan II | 0.853 | **0.914** | 0.742 | 0.707 |
| hold-out test | Human MIDOG22 | Breast/Ham XR | 0.755 | **0.760** | 0.64 | 0.463 |
| | | Breast/Ham S360 | 0.741 | **0.742** | 0.582 | 0.438 |
| | | Breast/Aperio CS2 | 0.764 | **0.783** | 0.623 | 0.458 |
| | | Neuroendocrine/ Ham XR | 0.626 | **0.699** | N/A | N/A |
| | Canine MIDOG22 | Lymphoma/3DHist Scan II | 0.753 | **0.804** | N/A | N/A |
| | | Cutaneous Mast Cell/ Aperio CS2 | 0.824 | **0.859** | N/A | N/A |
| | | Lung / 3DHist Scan II | 0.684 | **0.708** | N/A | N/A |
| | Aggregate MIDOG22 | All | 0.763 | **0.801** | 0.615 | 0.45 |

Table 3: Detection performance of our pipeline for mitotic figure (MF) detection and subtyping across different subgroups. Subtyping performance was evaluated assuming perfect MF recognition. Subtyping helps MF recognition in each given condition.

on CMC, CMC Rescan, our multi-tumor dataset and MIDOG22 (see Table 3). Moreover, we find that AMF classification is significantly more challenging for our model, as shown in the third and fourth column of Table 3. Note that the evaluations given in this table assume perfect recognition of MF and only evaluate the subtask of AMF classification. Our evaluation of the precision-recall-curves in the supplementary Figure S2 reveals a similar performance across scanners for breast cancer, demonstrating the scanner-robustness of our scheme, a notably high performance for the MF recognition in mast cell tumor, and a deterioration of performance on the neuroendocrine and lung cancer tumors. Lastly, as shown in the supplemantary Figure S1, there is a strong correlation between the predicted MC and the respective ground truth, and a less significant correlation for the AMF/NMF ratio.

When predicting on the TCGA-BRCA dataset, we found an average count of mitoses per ROIs of 26.67 and an average count of atypical mitoses within the ROI of 1.01. The AMF/NMF ratio on WSIs had a mean value of 0.10, whereas the same metric, when evaluated on the ROIs of the MC was 0.13, indicating a slightly higher rate of AMFs within the ROIs. The survival prediction, shown in Figure 3, shows that the MC stratified survival into two groups. The Chi-squared test indicates significant difference ($p < 0.01$) between the groups. On the other hand, even for an optimized cutoff value of 0.11, the test did not indicate significant differences between the groups stratified by the AMF/NMF ratio. We found the AMF/NMF ratio calculated on the hotspot ROI did not stratify survival at all.

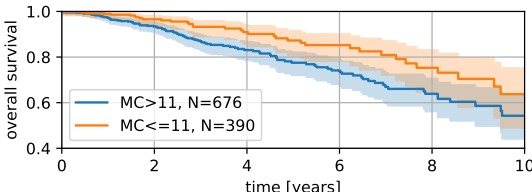 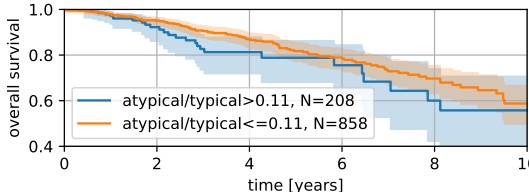

Figure 3: Kaplan Meier curves for the 10 year overall survival of breast cancer patients of the BRCA-TCGA dataset, for the most discriminating cutoff value. MC (left plot) discriminates subgroups significantly ($p < 0.01$), while AMF/NMF ratio-based stratification (right plot) is non-significant ($p = 0.10$)

## 4. Discussion and Conclusion

Our work shows that by using animal histopathology data, we can train a highly competitive MF detector for human breast cancer. The training scheme, utilizing registered rescans of the glass slides, incorporated a high degree of scanner robustness, as our results show. While the AMF detection itself yielded only mediocre classification rates, likely caused by the high inter-rater variability and the difficulty of the problem, it consistently regularlized the model and improved performance on the primary MF detection task.

We note that although the training paradigm introduced in our work reduced the number of false annotations in rescanned slides, it did not aim to recover MFs that were not annotated in the primary dataset, thus increasing the risk of false negatives in the rescanned slides. Moreover, the measurable effect of rescan filtering depends on how often the rescans are sampled during training. The use of immunohistochemistry in a restaining procedure could help in the identification of OOF mitoses in future work (Tellez et al., 2018).

The prediction of survival on the external BRCA dataset underscores the robustness of our pipeline. While we chose the threshold for subgroup separation in the survival analysis post-hoc, the value of 11 mitoses per $2\,mm^2$ is well within expectations given the current CAP guidelines (Fitzgibbons and Connolly, 2023). In contrast to the findings of Lashen et al. (Lashen et al., 2022), we did not find the AMF/NMF ratio to be significantly stratifying for overall survival, even for an optimized threshold value. While this could be linked to the performance of our classifier, we also observe a striking difference in the apparent perception of mitoses being atypical, which is expressed by a mean AMF/NMF ratio of 0.2 on TCGA-BRCA and a mean count of 2 AMFs per ROI in the original work (Lashen et al., 2022), where our overall estimates are significantly lower.

In conclusion, we demonstrated that our dual-pronged approach of using rescanned slides and MF subtyping allowed us to train a highly competitive MF detection approach using animal data only, as benchmarked on the multi-species MIDOG22 dataset. We envision that future improvements to the AMF subtyping task will allow us to find statistically significant and clinically meaningful uses for detecting atypical mitoses.

## Acknowledgments

M.A. acknowledges support by the German Research Foundation (project number 520330054). C.A.B. acknowledges funding by the Austrian Science Fund (FWF, project number: I 6555).

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

## Appendix A. Additional performance evaluations

In Figure S1 we show correlations and scatter plots of the ground truth mitotic density and AMF/NMF ratios versus our predicted ones on the MIDOG22 dataset. For both, we show the correlation for three different decision thresholds to demonstrate the overall robustness of the correlation. In each case, the second label showing the orange scatter plot show performance for the optimal threshold on the validation set, while the blue and green show a lower rand higher threshold, respectively.

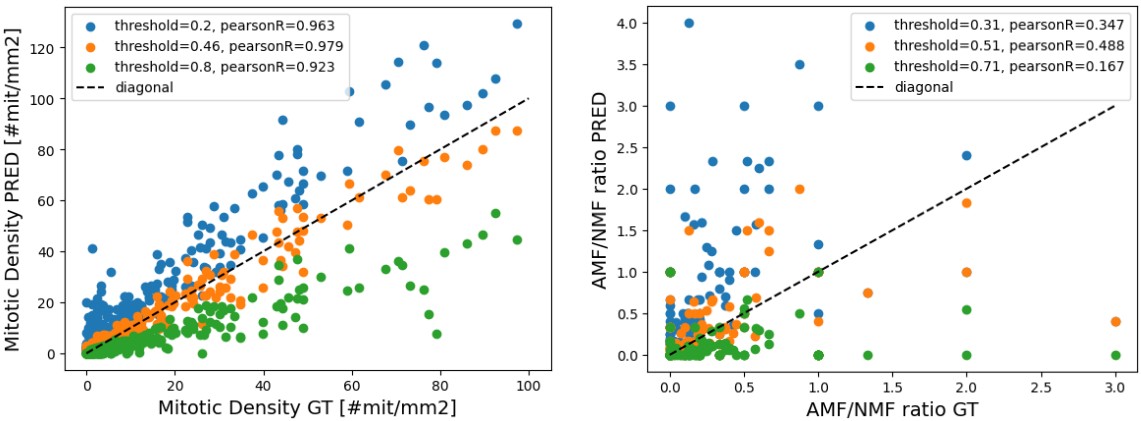

Figure S1: Scatter plots showing correlation between ground truth and estimated mitotic density (left) and AMF/NMF ratios (right) for each ROI of the MIDOG22 dataset.

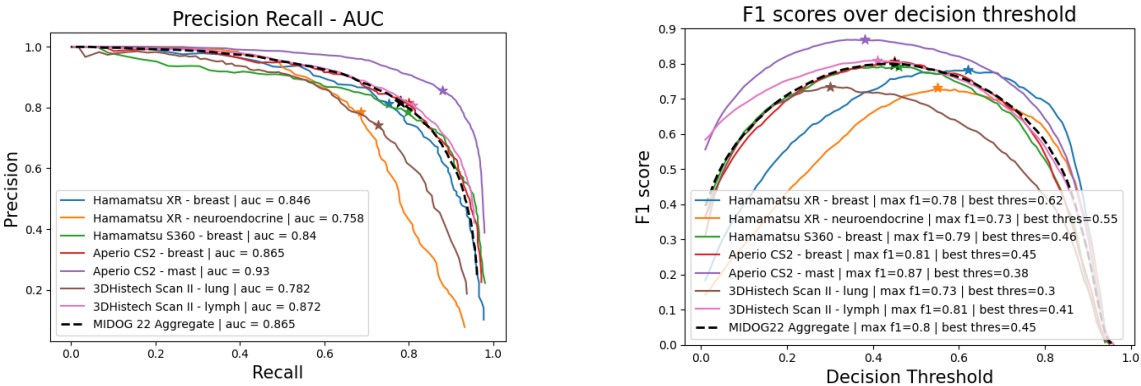

Figure S2: Quantitative results of best model on MIDOG22 dataset by scanner/organ subgroups. Left: PR-AUC. Right: F1-scores over decision thresholds.

## Appendix B. Multi-Tumor Dataset Composition

Our in-house multi-tumor dataset consists of 17 different cancer types. We selected one ROI per WSI spread over 156 WSI, spread over the cancer subtypes. We show the precise dataset composition in Table S1.

| Tumor Type | #ROI | #MF | #AMF |
|---|---|---|---|
| AdrencorticalTumors | 2 | 5 | 2 |
| ColonCarcinoma | 10 | 365 | 12 |
| GastrointestinalStromalTumors | 9 | 139 | 26 |
| HemangioSarcoma | 9 | 290 | 9 |
| HepaticCarcinoma | 8 | 33 | 3 |
| Lymphoma | 10 | 672 | 5 |
| MammaryCarcinoma | 8 | 363 | 12 |
| MastCellTumor | 12 | 392 | 31 |
| Melanoma | 11 | 593 | 61 |
| Meningioma | 6 | 28 | 4 |
| OsteoSarcoma | 15 | 434 | 21 |
| Pheochromocytoma | 7 | 104 | 3 |
| ProstateCarcinoma | 4 | 74 | 13 |
| PulmonaryCarcinoma | 10 | 329 | 67 |
| RenalCarcinoma | 7 | 233 | 10 |
| SoftTissueSarcoma | 13 | 170 | 10 |
| UrothelialCellCarcinoma | 9 | 371 | 35 |

Table S1: Overview of the number of ROIs and MFs annotated per tumor type in our multi-cancer training dataset.

## Author contributions

R.H.J.F. led the study design and data acquisition, trained and evaluated the models, and spearheaded the algorithm development. M.A. performed the survival analysis and contributed to algorithm development. C.A.B. provided annotations for the multi-tumor dataset and the mitotic phases. All others jointly wrote the manuscript.

