# OpenReview forum: "Improving CNN-Based Mitosis Detection through Rescanning Annotated Glass Slides and Atypical Mitosis Subtyping"
_MIDL.io/2024/Conference — MIDL 2024 Poster_

### Official Review · Reviewer_gJtB · 2024-02-29

**Confidence:** 4
**Preliminary Rating:** 5
**Recommendation:** Oral
**Final Rating:** 5

**Summary:**

In this paper, the authors develop a deep learning detection and classification of mitotic figures (MF). They create a dataset with the goal of robustness against organs/species, scanners, and WSI (robust against bias in ROI selection). The detection pipeline consists of a detector to propose MF candidates and a classifier to accept or reject the proposals. The main focus of the paper is on the classifier and the data. The authors show that the classifier can be consistently improved when training it in a multi-task-learning (MTL) fashion together with classification of normal (NMF) vs atypical (AMF) MF. Further, survival stratification is predicted using either the number of MF in a hotspot region or the NMF vs AMF ratio. However, the latter approach did not yield significant findings.

**Strengths:**

* the paper is well organized and structured, easy to follow
* The contributions of the paper are clear and the experiments are suited to evaluate the performance of the approach
* The topic is relevant and the results are promising
* Robustness is an important issue and is addressed in three ways
* In is interesting to see both the results for "Perfect Primary" and "End2End" in Table 3, as this reveals the effect of the conditional chain in the AI pipeline

**Weaknesses:**

* cut-off is chosen post-hoc. Instead, the threshold could have been optimized using the raining or validation dataset
* Also, introducing this threshold may introduce a bias as a e.g. certain type of AMF could be more difficult to predict and is thus more likely to stay below the set threshold
* The discussion contains a statement about why MFs were not tried to be recovered on the original scan and that the thresholds were set post-hoc. This is both important information that should have been positioned in sections 2.2 and 2.4, respectively

**Detailed Comments:**

* In table 1 the resolution of the resulting images could be added to see whether resolution differences affect the models' performance
* In section 2.2, it says "We show examples of false positive cases in Figure 2", probably "false negatives" is meant here

**Justification Of Final Rating:**

The paper addresses the relevant issue of robustness by creating a very diverse dataset. It also once again shows that MTL can improve DL models. The paper is stringently structured, clearly written and easy to follow.

**Justification Of The Preliminary Rating:**

The paper addresses the relevant issue of robustness by creating a very diverse dataset. It also once again shows that MTL can improve DL models. The paper is stringently structured, clearly written and  easy to follow.

**Questions To Address In The Rebuttal:**

* It is not really clear how the removal of uncertain MF classifications as described in section 2.2 affects the training of the classifiers. The authors should comment on this and why they think that removing uncertain MFs does not reduce the significance or their results.
* The authors should include a statement why they do not balance NMF vs AMF
* would be very interesting to see whether an optimization of the thresholds using training and/or validation data would result in similar values and how this affects the accuracy of the ratios
* Figure 3b is not described in the text

**Special Issue:**

No

---

> ### Author Response · Authors · 2024-03-14
> **Our response to the review**
>
> We appreciate the in-depth and very supportive comments of the reviewer and we have revised our manuscript according to the comments, which, we hope, will find the reviewer’s appreciation. Here are our replies to the points raised by the reviewer:
>
> > cut-off is chosen post-hoc. Instead, the threshold could have been optimized using the raining or validation dataset
>
> We apologize for the misconception that we determined the cut-off values post-hoc for the mitotic figure detection, which might have been caused by presenting the F1 score dependency on the cutoff for each subgroup in Figure 3. However, it's important to clarify that we did, indeed, optimize the cutoff based on the validation set, a point we elaborate on in the revised document.
>
> Regarding the survival prediction on TCGA, we did employ a post-hoc determination of an optimal subgroup-dividing threshold. Such an approach aligns well with other comparable recent works [1,2,3]. Subdividing the TCGA data for the specific purpose of defining and validating a cut-off would not only impair the comparability to other studies, but also severely reduce the statistical power of the sample, which is crucial for survival analysis. We, therefore, chose not to take this route.
>
> [1] 10.1038/s41379-022-01080-0
>
> [2] 10.21203/rs.3.rs-2408990/v1
>
> [3] 10.1038/s41598-023-40042-7
>
> > Also, introducing this threshold may introduce a bias as a e.g. certain type of AMF could be more difficult to predict and is thus more likely to stay below the set threshold
>
> As with any detection task employing subgroups, we can expect a dependency of the optimal threshold on the subgroup. However, analysis of this is, in our view, out of scope for the current work, since we did not employ a subclassification of AMFs in our pipeline in this work. To clarify, we also exchanged Figure 1, which might have given rise to this impression.
>
> > The discussion contains a statement about why MFs were not tried to be recovered on the original scan and that the thresholds were set post-hoc. This is both important information that should have been positioned in sections 2.2 and 2.4, respectively
>
> We acknowledge your point, and appreciate your suggestion on the placement of these critical pieces of information. Still, we feel it is important to clarify, once again, as stated earlier, that the cutoff for mitosis detection was not a post-hoc decision. We have revised the manuscript to underscore this in section 2.2. As for the statement regarding survival analysis, we reformulated this in section 2.4 for more clarity.
>
> > In table 1 the resolution of the resulting images could be added to see whether resolution differences affect the models' performance
>
> Thanks for this comment. We added this information to Table 1. We resampled all images to 0.25 microns/px for processing, which we also now added to the manuscript.
>
> > In section 2.2, it says "We show examples of false positive cases in Figure 2", probably "false negatives" is meant here
>
> We acknowledge that the term “false positive” might be misleading for an annotation. Hence, we switched our terminology to “false annotations” and “missing annotations”. Figure 2 shows false annotations, since the annotation exists but the object is no longer recognizable as a mitotic figure in the rescan.
>
> > It is not really clear how the removal of uncertain MF classifications as described in section 2.2 affects the training of the classifiers. The authors should comment on this and why they think that removing uncertain MFs does not reduce the significance or their results.
>
> We appreciate this comment and ran an ablation study on this question, which the reviewer will find in the revised manuscript in Table 2. In fact, as the results show, the filtering approach does improve the classifier performance consistently on all of the evaluated models. We should note that, given the sampling scheme only samples 25% from the rescanned slides, the effect would likely be even more pronounced when only training on the rescans.
>
> > The authors should include a statement why they do not balance NMF vs AMF
>
> The reviewer is right and ideally you would go even further to sample the mitotic subtypes and phases equally. However, we found in preliminary experiments that this did not benefit the overall mitotic detection performance. We hypothesize that our support for AMF in the dataset is still not big enough over all domains such that the model does not overtrain on the few ones available in some domains. We will explore this in future work with further dataset extensions.
>
> > Figure 3b is not described in the text
>
> Thanks for pointing this out. Ultimately, we decided to move Figure 3b to the supplementary material, as also the peer review required us to make space for additions requested by the reviewers.

---

### Official Review · Reviewer_R5oF · 2024-02-29

**Confidence:** 4
**Preliminary Rating:** 4
**Final Rating:** 5

**Summary:**

The paper introduces a pipeline for detecting and subclassifying mitotic figures (MFs) in histopathological images, particularly focusing on canine and human breast cancer datasets. It highlights the challenges in accurate MF identification and proposes solutions such as rescanning slides and incorporating mitosis subtyping. The proposed pipeline demonstrates improved performance in MF detection and subtyping, as well as survival prediction in breast cancer patients.

**Strengths:**

1. The paper addresses various challenges in MF detection, including morphological complexities and inter-rater variability, by proposing a multi-task learning pipeline.
2. The study incorporates datasets from different scanners, organs, and species.
3. The study proposes a training pipeline for rescanned slides, to address artifact issues in MF detection.
4. Incorporating mitosis subtyping leads to performance improvements.
5. The authors also attempt to use the AMF to NMF ratio computed via the suggested models to predict survival in breast cancer patients.

**Weaknesses:**

In the submission, the authors do not report plans on making data or code public. Open-sourcing the code with weights/It data would strongly increase the value of this work to the digital pathology research community.

**Detailed Comments:**

In the abstract, the 0.80 value is reported without a metric specified. I assume it is F1 score?

**Justification Of Final Rating:**

It is a good-quality study. The paper is well-structured, easy to follow, and addresses a relevant issue. The authors addressed my comments. Based on these inputs, the preliminary rating is preserved (Weak accept).

**Justification Of The Preliminary Rating:**

It is an interesting study, that addresses a relevant topic. The study is well-executed. However, I hope the authors are planning to share their code, weights, and data, as this would increase the value of this work to the research community.

**Questions To Address In The Rebuttal:**

Points mentioned in Weaknesses and Detailed Comments.

---

> ### Author Response · Authors · 2024-03-14
> **Our comments to the review**
>
> We appreciate the constructive comments of the reviewer and we have revised our manuscript according to the comments, which, we hope, will find the reviewer’s appreciation. These are our responses to the points raised by the reviewer:
>
> > In the submission, the authors do not report plans on making data or code public. Open-sourcing the code with weights/It data would strongly increase the value of this work to the digital pathology research community.
>
> We are preparing the release of the dataset, however, since the dataself is of substantial size and complexity and the publication warrants an in-depth description, it is not possible within the review time span. Additionally, at this point alterations to the co-authors list for this paper are no longer feasible. There were several other researchers who played a role in data preparation, and we firmly believe in appropriately acknowledging all contributions in scientific endeavors. Hence, we feel it's crucial that all parties involved receive proper recognition.
>
> Regarding the release of the detector source code: As reviewer 3zQm pointed out, there is no new model architecture involved, and our paper is based on well-known application of object detectors, so we see no added value in releasing yet another very similar pipeline to the suite of already available code bases for this purpose, given the pace of software and library development in our field, would soon be outdated.
>
> > In the abstract, the 0.80 value is reported without a metric specified. I assume it is F1 score?
>
> We are sorry for not stating this clearly. The reviewer is correct: It is the F1 score, we have fixed this in the revised abstract.

---

### Official Review · Reviewer_3zQm · 2024-03-02

**Confidence:** 4
**Preliminary Rating:** 2
**Final Rating:** 4

**Summary:**

The article proposes an approach for identification of mitotic figures in WSIs. The contributions are two folds: a composite dataset and a training pipeline. The composite dataset is collected from three different sources: two public and one in-house. The training pipeline includes the preprocessing of the dataset: annotations from the original WSI are transferred to the rescans using a WSI-level registration algorithm and a filtering approach to remove the ambiguity. The filtering approach is based on ensembling of 10 different classification architectures which are used to mask the uncertain annotations based on the threshold. Mitosis detection pipeline consists of a detection network and two classification networks (an additional loss is used for classification: atypical MF subtyping).

**Strengths:**

The paper has created a large dataset by combining different datasets and aspects to induce robustness. Including secondary loss function have also improved the classification performance. The writing is detailed with required references.

**Weaknesses:**

1. Writing:
+ The writing can be improved by including a graphical abstract summarizing the overall pipeline. It will improve the readability of the paper.

2. Novelty and Results:
+ The novelty aspect of the paper is very limited.
     + The major contributions come from the aggregation of the dataset and training process.
     + The architecture and other algorithms (WSI registration) are utilized from the existing work (Li et al., 2018; Piansaddhayanaon et al., 2023, Marzahl et al., 2021).
+ Authors have proposed a filtering approach to improve the annotation.
     + However, there is no ablation study to highlight its significance.
+ There is no comparison with the existing approaches:
     * The existing methods should also be trained with the new composite dataset for the fair comparison

**Detailed Comments:**

Please refer to *weaknesses* section above.

**Justification Of Final Rating:**

The authors have provided some relevant arguments related to the raised concerns. They have also included the requested changes. Based on these inputs, the preliminary rating is revised in favour of acceptance.

**Justification Of The Preliminary Rating:**

The decision is based on the novelty and results aspects. The novelty is limited and the proposed pipeline is adopted from the existing work. The presented results also not cover all the aspects of the methodology. Please refer to *weaknesses* section.

**Questions To Address In The Rebuttal:**

+ The ablation study showing the impact of filtering in data preprocessing
+ Comparison with existing methods (trained with proposed dataset)
+ Do authors plan to release the dataset?

---

> ### Author Response · Authors · 2024-03-14
> **Response to the review**
>
> We appreciate the constructive comments of the reviewer and we have revised our manuscript according to the comments, which, we hope, will find the reviewer’s appreciation. Here's our reply to the points raised by the reviewer:
>
> > The writing can be improved by including a graphical abstract summarizing the overall pipeline. It will improve the readability of the paper.
>
> We agree with the reviewer on this, and replaced the previous Figure 1 with a completely new Figure showcasing the overview of our method.
>
>
> > The novelty aspect of the paper is very limited. The major contributions come from the aggregation of the dataset and training process. The architecture and other algorithms (WSI registration) are utilized from the existing work (Li et al., 2018; Piansaddhayanaon et al., 2023, Marzahl et al., 2021).
>
> We view this as an asset, rather than a liability of our paper: We utilize our comprehensive understanding of the issue at hand to engineer a robust solution, applicable to a vast number and variety of architectures. Our approach enhances and complements existing research in the domain. Reusing glass slides to embed characteristics of different scanners into a dataset was never done before by any other group, despite the advantages being obvious and clearly shown by our work. As our results demonstrate, the efficiency of a pipeline hinges substantially (if not more so) on the data loading strategy and dataset attributes rather than merely on the model architecture.
>
>
> > Authors have proposed a filtering approach to improve the annotation. However, there is no ablation study to highlight its significance.
>
> We agree with the reviewer and provide the ablation study as an additional column of Table 2 in the revised manuscript.
>
> > There is no comparison with the existing approaches. The existing methods should also be trained with the new composite dataset for the fair comparison
>
> We respectfully disagree with the reviewer on this point. Our paper does not primarily present a new architecture, making model comparisons outside the scope of our paper. We draw attention to Table 2 in our paper where we have indeed offered a comparative study of three cutting-edge classification architectures - DenseNet, EfficientNet and ResNet - in relation to our task, only to find negligible variation in the results. Given our candidate detection task is undertaken with a set operating point targeting a 99% recall rate, virtually any state-of-the-art object detection would suffice. Hence, we respectfully argue that an additional comparison at this stage of the architecture would not contribute any new insights to our paper.
>
> > Do authors plan to release the dataset?
>
> We are preparing the release of the dataset, however, since the dataself is of substantial size and complexity and the publication warrants an in-depth description, it is not possible within the review time span. Additionally, at this point alterations to the co-authors list for this paper are no longer feasible. There were several other researchers who played a role in data preparation, and we firmly believe in appropriately acknowledging all contributions in scientific endeavors. Hence, we feel it's crucial that all parties involved receive proper recognition.

---

### Author Response · Authors · 2024-03-14
**Overall comments on the review**

We sincerely thank all three of the reviewers for providing insightful and constructive criticism of our paper. We think that the new, revised manuscript has sincerely benefitted from the review. The main points that we updated were:
- We provide a new main Figure, serving as graphical abstract, as requested by Reviewer 3zQm.
- We ran an ablation study on the automatic removal of uncertain mitotic figures, as requested by reviewers 3zQm and gJtB.
- We improved the overall phraseology according to the reviewers’ comments.

Please find the detailed comments in the responses to each reviewer.

---

### Author Response · Authors · 2024-03-28
**End of review period**

Dear area chair,

We appreciate the constructive feedback and positive evaluation of our paper. Regrettably, none of the reviewers have actively participated in the discussion phase. Nevertheless, we are confident that we have addressed all the issues raised in the initial review. We hope the lack of engagement as an indication that the reviewers are in agreement with our responses.

Warm regards,

Marc Aubreville (on behalf of the authors)

---

### Meta-Review · Area_Chair_4NJ8 · 2024-04-04

**Recommendation:** Accept (Poster)
**Confidence:** 4

**Metareview:**

The paper presents a pipeline for detecting and subclassifying mitotic figures (MFs) in histopathological images, with a focus on canine and human breast cancer datasets. It introduces a composite dataset and a training pipeline that includes preprocessing and a filtering approach to improve annotation quality. While the paper demonstrates improvements in MF detection and subtyping, reviewers highlight several weaknesses, including limited novelty, lack of comparison with existing approaches, and insufficient clarity in certain aspects.

While the paper presents a comprehensive pipeline for MF detection and subtyping, addressing the weaknesses highlighted by reviewers could significantly strengthen its contribution and impact. Suggestions include conducting an ablation study for the filtering approach, comparing with existing methods, and clarifying certain aspects of the methodology and results presentation.

---

### Decision · Program_Chairs · 2024-04-06

Accept (Poster)